# Amplifying Prominent Representations in Multimodal Learning via Variational Dirichlet Process

**Tsai Hor Chan**[*]
University of Pennsylvania
Tsaihor.Chan@PennMedicine.upenn.edu

**Feng Wu**[*]
University of Hong Kong
fengwu96@connect.hku.hk

**Yihang Chen**
University of Hong Kong
yihangc@connect.hku.hk

**Guosheng Yin**
University of Hong Kong
gyin@hku.hk

**Lequan Yu**[†]
University of Hong Kong
lqyu@hku.hk

## Abstract

Developing effective multimodal fusion approaches has become increasingly essential in many real-world scenarios, such as health care and finance. The key challenge is how to preserve the feature expressiveness in each modality while learning cross-modal interactions. Previous approaches primarily focus on the cross-modal alignment, while over-emphasis on the alignment of marginal distributions of modalities may impose excess regularization and obstruct meaningful representations within each modality. The Dirichlet process (DP) mixture model is a powerful Bayesian non-parametric method that can amplify the most prominent features by its richer-gets-richer property, which allocates increasing weights to them. Inspired by this unique characteristic of DP, we propose a new DP-driven multimodal learning framework that automatically achieves an optimal balance between prominent intra-modal representation learning and cross-modal alignment. Specifically, we assume that each modality follows a mixture of multivariate Gaussian distributions and further adopt DP to calculate the mixture weights for all the components. This paradigm allows DP to dynamically allocate the contributions of features and select the most prominent ones, leveraging its richer-gets-richer property, thus facilitating multimodal feature fusion. Extensive experiments on several multimodal datasets demonstrate the superior performance of our model over other competitors. Ablation analysis further validates the effectiveness of DP in aligning modality distributions and its robustness to changes in key hyperparameters. Code is anonymously available at https://github.com/HKU-MedAI/DPMM.git

## 1 Introduction

Multimodal learning aims to aggregate information from multiple modalities to generate meaningful representations for downstream tasks. It has been widely explored in the context of vision-language

---

[*]Equal contribution.
[†]Corresponding author.

39th Conference on Neural Information Processing Systems (NeurIPS 2025).

models [11, 9], audio-visual applications [5, 35, 20], image-video models [13, 12] and healthcare applications [50, 18]. For example, multimodal learning has been applied to various healthcare tasks such as clinical outcome prediction [61, 50], report generation [42, 4], and clinical trial site selection [45]. One of the main tasks in multimodal learning is to fuse information from different modalities, which can be divided into early, joint, or late fusion strategies [21]. The joint fusion strategy is the most popular multimodal fusion paradigm [21] due to its strong ability to capture structural inter-modality interactions. It aims to align the distributions of different modality representations, creating a unified representation that effectively captures these interactions and preserves meaningful cross-modal information for downstream prediction tasks [18].

However, due to the inherent heterogeneity of different modalities (e.g., medical images, medical reports, electronic health records), properly aligning the distributions of various modalities for fusion remains a challenging problem. Despite the success of existing alignment strategies, such as contrastive learning [47] and adversarial alignment [31], most of them primarily focus on aligning features while neglecting the important intra-modality representations within each modality. As a result, this leads to weaker fused representations, ultimately limiting the performance of downstream tasks and reducing the generalizability and robustness of multimodality models. Consequently, there is a strong need for a novel approach that can not only effectively align the feature distributions but also preserve the important intra-modal representations.

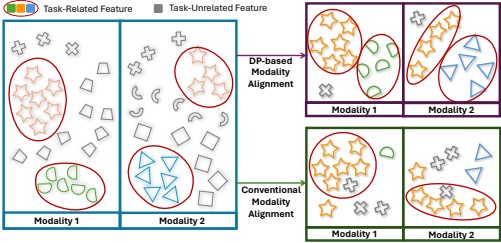

Figure 1: Dirichlet process based modality alignment (top right) vs. conventional modality alignment (bottom right). The Dirichlet process allocates more weights to capture the prominent features (e.g., $\star$, $\triangle$) while conventional methods inevitably shrink the intra-modal features during alignment.

Bayesian nonparametric methods have shown great success in representation learning due to their flexibility and capability in density estimation. Specifically, the Dirichlet process (DP) has been frequently adopted since it can highlight the most significant features in the covariates with its richer-gets-richer property [60]. Figure 1 presents an illustrative example of how DP can amplify the feature signals for each modality. Despite its capability, the potential of DP in tackling multimodal representation learning is still under-explored. Moreover, the issue of unpaired observations (i.e., missing observations on some modalities) presents a significant challenge in multimodal learning. Most existing methods operate under the assumption that all modalities are available for every observation. However, in real-world scenarios, it is common for certain modalities, such as medical images or reports, to be absent for some patients due to clinical or administrative reasons in health care [54]. The existing solutions either discard these observations or impute simple values [18] (e.g., zeros) to address the missing modality problem. However, the feature signals in each modality after learning the alignment objectives are shrunk, leading to weaker feature representations and worse imputation performance. Therefore, it is necessary to amplify the intra-modal representations to generate representative features for the observations with unpaired observations.

In light of the above challenges, we propose a novel Dirichlet process-driven multimodal learning framework, namely DPMM (**D**irichlet **P**rocess **M**ixture **M**ultimodal Learning), to tackle the multimodal fusion paradigm from a probabilistic perspective. We formulate the multimodal feature distribution as a Gaussian mixture model whose weights are allocated by DP, such that prominent features can receive increasing attention. Our contributions can be summarized as: (1) We introduce a novel multimodal learning framework via the Dirichlet process to amplify the feature signals while properly aligning them across modalities. We treat the joint distribution as a DP mixture model to allow DP to select the most useful features using its richer-gets-richer property. (2) We adopt stochastic variational inference to efficiently optimize the proposed Dirichlet process model, which overcomes the scalability limitations of traditional optimization algorithms based on Markov chain Monte Carlo (MCMC) and enables the scalability of our model to large-scale datasets. (3) We utilize the learned marginal distributions as the representation generator to accurately impute the missing observations with the prominent features highlighted by DP. This enhances the features supplied to training and downstream tasks and hence leads to improved performance. (4) Empirical results on four

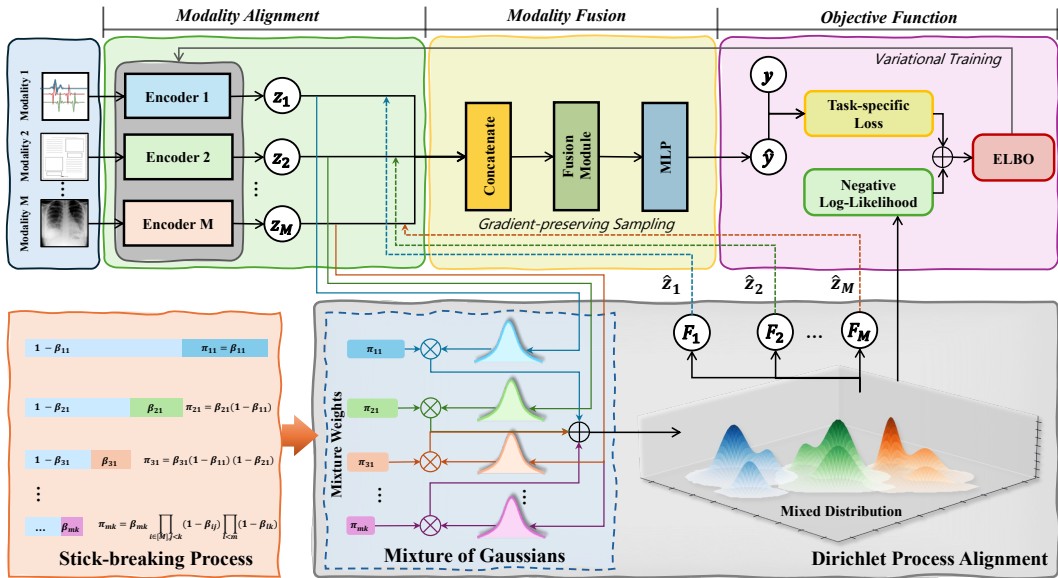

Figure 2: Overview of our proposed DPMM framework. We first input the data to the encoders to obtain latent embeddings $z_1, \ldots, z_M$. We then compute the mixture weights $\{\pi_{mk} : m \in [M], k \in [K]\}$ using the stick-breaking formulation of the DP. The weighted likelihood is then computed with the mixture weights and the multivariate Gaussian mixture assumption. Finally, by integrating the task-specific loss, the evidence lower bound (ELBO) for training the encoders can be obtained. To handle unpaired samples during training, we adopt gradient-preserving sampling which can impute the missing observations while the gradient can still be backpropagated.

multimodal datasets demonstrate the superior performance of our framework and ablation analysis further corroborates the effectiveness of DP in modality alignment and feature amplification.

## 2 Related Works

**Cross-modal Alignment.** Cross-modal alignment aims to make representations of different modalities comparable and aligned in a shared latent space [22, 52], and it has been widely applied in various multimodal tasks, such as image-text retrieval [42], image captioning [63], and multimodal classification [10]. Some existing works focus on learning a shared representation using fusion strategies, such as concatenation [18, 46] or transformer-based mechanisms [61, 45] without alignment. Despite being effective for certain tasks, these methods may yield suboptimal results when dealing with highly heterogeneous modalities [54]. To address this issue, some works use contrastive learning to align representations by maximizing the similarity between different modalities [47, 48]. Other studies use variational inference methods [44, 62, 10], such as variational autoencoders (VAEs), to align representations by minimizing the Kullback–Leibler (KL) divergence between latent distributions of different modalities. However, these methods often emphasize on the shared information between modalities, which may compromise the preservation of crucial modality-specific features that are necessary for downstream tasks.

**Deep Bayesian Nonparametric Methods.** Bayesian nonparametric methods, including Gaussian process (GP) [40, 30, 16, 27, 41], Dirichlet process (DP) [8, 60, 23, 2], and Pólya tree [34, 36], have shown significant success in parameter estimation as they perform searching in an infinite dimensional parameter space. Among these methods, DP has been extensively applied in clustering and density estimation for its richer-gets-richer property. In light of the rapid development of deep learning, many works [43, 2, 16] introduce variational inference to render traditional Bayesian nonparametric methods to a large scale. However, the capability of deep DP in multimodal learning is still undervalued. Recently, there are some works applying DP to multimodal learning [43, 53]. For example, MoM-HDP [53] is a multimodal topic model via a hierarchical Dirichlet process which

extends its latent Dirichlet allocation (LDA) counterparts [3, 1]. However, these methods are limited to topic modelling problems, where their assumptions are difficult to be generalized to other scenarios.

**Learning with Missing Data.** Traditional multimodal learning assumes all modalities are available, while in reality, some observations may be missing, e.g., missing medical images or reports in clinical data. Late fusion is a common strategy to address missing modalities by aggregating predictions [56] or latent space representations [45] from the available modalities. Despite its effectiveness, late fusion treats each modality independently and lacks interactions between them. Some existing works focus on extracting shared information across modalities [6, 54, 18]. However, learning such shared representations can be challenging, particularly when the modalities are highly different in distributions (e.g., tabular data and image data). Many approaches attempt to preserve model performance via modelling the relationships between them [59, 50] or generating a global representation for the missing data [18]. Other methods assume that the missing modality follows a certain distribution, imputing the missing values using the mean or mode of that distribution [33]. Despite their successes, the distributional assumptions emphasize more on the structural interactions between the modalities, while highlighting the prominent features or signals is less considered. The resulting feature generator would produce weaker signals for missing modalities, which leads to less representative features.

## 3 Methodology

### 3.1 Preliminaries

**Dirichlet Process.** Denoted as $\mathrm{DP}(\eta, G)$, the Dirichlet process is a random probability measure on the sample space $\mathcal{X}$, such that for any measurable finite partition $S$ of $\mathcal{X}$, denoted as $S = \{B_i\}_{i=1}^K$,

$$(G(B_1), G(B_2), \ldots, G(B_K)) \sim \mathrm{Dir}(\eta G(B_1), \eta G(B_2) \ldots, \eta G(B_K)),$$

where $G$ is the base probability measure, $\eta$ is the concentration parameter and $\mathrm{Dir}(\cdot)$ denotes the Dirichlet distribution.

**Multimodal Learning.** Given the multimodal training dataset $\mathcal{D}_{\mathrm{tr}} = \{(\boldsymbol{x}_1^{(i)}, \ldots, \boldsymbol{x}_M^{(i)}, y^{(i)})\}_{i=1}^n$, where $\boldsymbol{x}_m^{(i)}$ (for $m = 1, \ldots, M$) denotes the $i$-th observation of the $m$-th modality and $y^{(i)}$ is the corresponding label, the goal is to train a multimodal neural network $f_{\boldsymbol{\Theta}}(\cdot)$ with parameter $\boldsymbol{\Theta}$ such that the model achieves optimal performance in downstream tasks.

### 3.2 Multimodal Learning via Dirichlet Process

In our multimodal learning framework, we formulate the multimodal feature distribution as a Gaussian mixture model whose weights are allocated by DP. The DP allocation can ensure that prominent features receive increasing attention. An overview of the proposed DPMM is provided by Figure 2, and the detailed computation steps are given in Algorithm 1.

**Stick-breaking Process for Mixture Weights.** We define the Dirichlet process multimodal learning framework as a mixture of $M \times K$ components, where each mixture is assumed to be a multivariate Gaussian distribution. We adopt the popular stick-breaking process to formulate the prior distribution of the weights as follows,

$$\beta_{mk} \sim \mathrm{Beta}(1, \eta), \qquad \pi_{mk} = \beta_{mk} \prod_{i \in [M], j < k} (1 - \beta_{ij}) \prod_{l < m} (1 - \beta_{lk}), \tag{1}$$

where $\pi_{mk}$ is the probability assigned to the $k$-th mixture component of the $m$-th modality. Figure 3 provides a graphical illustration of the stick-breaking process under the multimodal context.

By adopting a stick-breaking process to select the mixture weights, the DP can automatically select the most prominent features among the modalities by its rich-gets-richer property.

**Gaussian Mixture as Marginal Distribution.** To enable a more flexible assumption of the feature representation at high dimensions, we assume the feature distribution of the $m$-th modality follows a

$K$-mixture of multivariate Gaussian mixture model (GMM),

$$f_m(\boldsymbol{z}) = \sum_{k=1}^{K} \pi_{mk} \mathcal{N}(\boldsymbol{z}|\boldsymbol{\mu}_{mk}, \boldsymbol{\Sigma}_{mk}), \quad m = 1, \dots, M, \tag{2}$$

where $\pi_{mk}$ is the mixture weight, $\boldsymbol{\mu}_{mk}$ is the mean vector, and $\boldsymbol{\Sigma}_{mk}$ is the covariance matrix of the $k$-th mixture component of the $m$-th modality. The GMM is a common technique in modelling high-dimensional distributions [29, 55, 8]. Let $\boldsymbol{\mu} = \{\boldsymbol{\mu}_{mk} : m \in [M], k \in [K]\}$ and $\boldsymbol{\Sigma} = \{\boldsymbol{\Sigma}_{mk} : m \in [M], k \in [K]\}$. We set $\boldsymbol{\mu}$ and $\boldsymbol{\Sigma}$ as learnable parameters to allow them to be updated by gradient back-propagation. We further let $\boldsymbol{\theta} = \{\boldsymbol{\mu}, \boldsymbol{\Sigma}\}$. For computational efficiency, we adopt the parameterization trick [8], which assumes each $\boldsymbol{\Sigma}_{mk}$ as a diagonal matrix.

With the density function for the $m$-th modality $f_m$ defined by (2), the joint distribution is

$$F(\boldsymbol{z}) = \sum_{k=1}^{K} \sum_{m=1}^{M} \pi_{mk} F_m(\boldsymbol{z}; \boldsymbol{\mu}_{mk}, \boldsymbol{\Sigma}_{mk}),$$

where the mixture weights are given in Eq. (1), and $F_m(\boldsymbol{z}; \boldsymbol{\mu}_{mk}, \boldsymbol{\Sigma}_{mk})$ is the distribution function of the $k$-th mixture of the $m$-th modality. Our probabilistic model in the plate notation is presented in Figure 4.

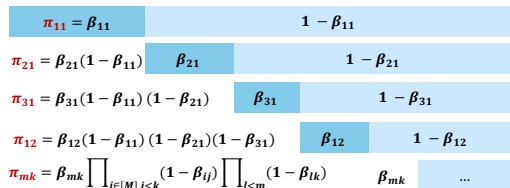

Figure 3: Illustration of the stick-breaking process under the multimodal context. We perform splitting by modality first and then by mixture components. This allows the DP to allocate weights to the most prominent features from each modality.

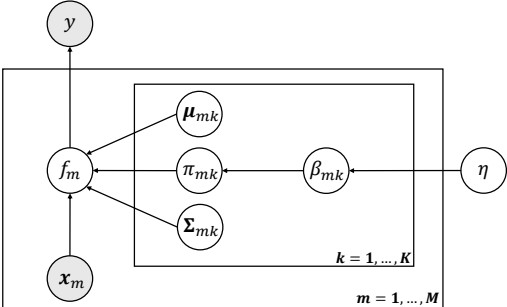

Figure 4: Plate notation of our proposed DPMM model.

**Discussion.** By modelling the multimodal feature distribution as a DP mixture model, the DP can allocate increasing attention to useful feature components $mk$. This highlights the most significant contribution of each modality $m$. On the other hand, the Gaussian components' covariances capture anisotropy (Mahalanobis geometry), aiding cross-modal structure learning. Hence our method can inherently optimize the tradeoff between highlighting prominent intra-modality features and effective cross-modal alignment.

**Expressiveness Under Truncation.** A DP can be truncated at $K$ without losing its expressiveness. Ishwaran and Zarepour [23] propose to represent a DP model with a finite mixture of models.

**Theorem 1.** *(Ishwaran and Zarepour [23]) Given the base distribution of a Dirichlet process $G$, for every measurable function $f$ integrable with respect to $G$, as $K \to \infty$, we have*

$$\int f(\boldsymbol{\theta}) dG^K(\boldsymbol{\theta}) \xrightarrow{\mathcal{D}} \int f(\boldsymbol{\theta}) dG(\boldsymbol{\theta}),$$

*where $G$ is approximated by an infinite mixture and $G^K$ is the DP mixture model with finite $K$ mixtures.*

One major insight from this theorem is the guaranteed convergence of the posterior distribution of the representation under the DP mixture. Even if we truncate the mixture at $K$, the learned distribution still preserves the property as if an infinite number of mixtures is used. The DP can best estimate the behaviour of embeddings in each modality and generate expressive features.

---

**Algorithm 1** Sampling algorithm of our proposed DPMM framework.

---

 **Input:** Multimodal encoder $f_\Theta(\cdot)$ with parameter set $\Theta$, concentration parameter $\eta$
1: Means and covariances of Gaussian distributions $\{(\boldsymbol{\mu}_{mk}, \boldsymbol{\Sigma}_{mk}), m = 1, \ldots, M, k = 1, \ldots, K\}$
2: Training set $\mathcal{D}_{\mathrm{tr}} = \{\boldsymbol{x}_1^{(i)}, \ldots, \boldsymbol{x}_M^{(i)}, y^{(i)}\}_{i=1}^n$
 **Output:** Trained $f_\Theta$
3: Initialize $\boldsymbol{\mu}_{mk} = \boldsymbol{0}$ and $\boldsymbol{\Sigma}_{mk} = \boldsymbol{I}$
4: **for** $(\boldsymbol{x}_1^{(i)}, \ldots, \boldsymbol{x}_M^{(i)}, y^{(i)})$ in $\mathcal{D}_{\mathrm{tr}}$ **do**
5:      Sample $\pi_{mk}$ with Eq. (1)
6:      $\hat{y}^{(i)} = f_\Theta(\boldsymbol{x}^{(i)})$, with $\boldsymbol{x}^{(i)} = (\boldsymbol{x}_1^{(i)}, \ldots, \boldsymbol{x}_M^{(i)})$            $\triangleright$ Forward propagation
7:      Compute task-specific loss $\mathcal{L}_{\mathrm{obj}}$ with $\hat{y}^{(i)}$ and $y^{(i)}$
8:      Compute the $\mathrm{KL}(q\|p)$ and hence the ELBO
9:      Back-propagate the ELBO to update $\Theta, \boldsymbol{\mu}, \boldsymbol{\Sigma}$
10: **end for**
11: **Return:** Trained $f_\Theta$

---

## 3.3 Stochastic Variational Inference

To render our method to a large scale, we adopt stochastic variational inference to estimate the parameters in the DP mixture. The variational family is specified as

$$q(\boldsymbol{\beta}, \boldsymbol{y}|\boldsymbol{\theta}) = \prod_{m=1}^M \prod_{k=1}^K q(\beta_{mk}) \prod_{i=1}^n q(y^{(i)}|\boldsymbol{\beta}, \boldsymbol{\theta}),$$

where $q(\cdot)$ is the variational posterior (an approximating distribution) chosen to be close to the intractable true posterior. The evidence lower bound (ELBO) is given by

$$\mathrm{ELBO} = -\mathrm{KL}(q(\boldsymbol{\beta})\|p(\boldsymbol{\beta})) - \mathrm{KL}(q(\boldsymbol{\theta})\|p(\boldsymbol{\theta})) - \sum_i \mathrm{KL}(q(y^{(i)})\|p(\hat{y}^{(i)})), \quad (3)$$

where $\mathrm{KL}(q(y^{(i)})\|p(\hat{y}^{(i)}))$ can be interpreted as the task-specific loss for the $i$-th sample. By adopting stochastic variational inference, we can treat the likelihood from the assumed probabilistic model as the surrogate loss for backpropagation, and hence the DP mixture model becomes feasible for multimodal learning in high dimensions.

**Variational updates of the posterior mixture weights.** We detail the closed-form updates for the variational posteriors associated with the stick-breaking weights. Let $\boldsymbol{\gamma}_{1,m}, \boldsymbol{\gamma}_{2,m} \in \mathbb{R}^K$ respectively denote the shape parameters of the Beta distributions in the variational factors $q(\beta_{mk}) = \mathrm{Beta}(\gamma_{1,mk}, \gamma_{2,mk})$ for modality $m$. Minimizing the KL terms yields

$$\gamma_{1,mk} = 1 + \sum_{i=1}^n \phi_{i,mk}, \qquad \gamma_{2,mk} = \eta + \sum_{i=1}^n \sum_{r=mk+1}^{MK} \phi_{i,r}, \quad (4)$$

where $\eta$ is the concentration rate, $\boldsymbol{\phi}_i \in \mathbb{R}^{MK}$ collects the posterior responsibilities (i.e., posterior unnormalized weights) over all $MK$ components for sample $i$. The log-responsibilities are

$$\log \phi_{i,mk} = \mathbb{E}_{\beta \sim q}\big[ \log \pi_{mk} \big] + \mathbb{E}\big[ \log p(\boldsymbol{x}^{(i)}) \big] + \mathbb{H}\big[ q_{\psi_{mk}}(\cdot \mid z_i = (m,k), \boldsymbol{x}^{(i)}) \big] + \mathrm{const}, \quad (5)$$

followed by normalization $\sum_{r=1}^{MK} \phi_{i,r} = 1$ for each $i$. Here, $\pi_{mk}$ are the mixture weights implied by the variational Beta factors, $\mathbb{H}$ is the entropy function, $z_i$ is the indicator function of a mixture component, $\boldsymbol{x}^{(i)} = (\boldsymbol{x}_1^{(i)}, \ldots, \boldsymbol{x}_M^{(i)})$ is the observed input, and $q_{\psi_{mk}}$ denotes the component-conditional variational distribution with parameters $\psi_{mk}$.

## 3.4 Missing Modality Imputation

Our DPMM can inherently impute the missing modalities as a probabilistic framework. While adopting the common missing at random (MAR) assumption, we obtain the marginal distributions $F_1, \ldots, F_M$ from the joint distribution $F$. For each $\boldsymbol{x}_m^{(i)}$ with missing observation on the $m$-th modality, we sample from the Gaussian mixture distribution, $\boldsymbol{x}_m^{(i)} \sim F_m$. In this way, the sampled embeddings contain both cross-modal information and prominent intra-modal information.

# 4 Experiments

## 4.1 Datasets and Experimental Settings

**Datasets and Preprocessing.** We evaluate the performance of our DPMM on two multimodal large-scale clinical datasets—MIMIC-III and MIMIC-IV, following the previous work [18]. As CXR images are not available in MIMIC-III, we replace them with clinical notes serving as the second modality. We extracted 25,071 ICU stays with EHR records from MIMIC-IV, 5,931 of which are matched to CXR images and reports. Similarly, we extracted 21,139 ICU stays with EHR records from MIMIC-III, with 5,273 stays matched to clinical notes. To evaluate the performance of DPMM on cross-modal alignment, we conduct experiments on *totally matched* bi-modal and tri-modal settings. We also evaluate the performance on *partially matched* datasets to demonstrate the robustness of DPMM in the presence of missing modalities.

Moreover, to show the generalization of our framework, we also conduct experiments on CMU-MOSI and POM datasets following the implementations of Liu et al. [32], which are general multimodal datasets encompassing video, audio, and language modalities. Table 1 provides an overview of the datasets used in our experiments. Further details on the datasets can be found in the supplementary materials.

Table 1: Summary of the real datasets.

| Dataset | No. Train | No. Valid | No. Test | No. Total |
|---|---|---|---|---|
| Complete Datasets | | | | |
| MIMIC-III | 14,681 | 3,222 | 3,236 | 21,139 |
| MIMIC-III NOTE | 3,652 | 815 | 806 | 5,273 |
| MIMIC-IV | 18,064 | 2,035 | 4,972 | 25,071 |
| MIMIC-CXR | 344,529 | 9,497 | 23,069 | 377,095 |
| CMU-MOSI | 1,283 | 229 | 686 | 2,198 |
| POM | 600 | 100 | 203 | 903 |
| Matched Datasets | | | | |
| MIMIC-III&NOTE | 3,652 | 815 | 806 | 5,273 |
| MIMIC-IV&CXR | 4,287 | 465 | 1,179 | 5,931 |
| MIMIC-IV&CXR&REPORT | 4,287 | 465 | 1,179 | 5,931 |

**Task & Evaluation Metrics.**
Following the common practice in clinical prediction tasks [18, 59, 50, 49], we focus on two clinical prediction tasks: (1) **In-Hospital Mortality (IHM)** prediction, which predicts whether a patient passes away during their hospital stay; and (2) **Readmission (READM)** prediction, which aims to predict whether a patient is readmitted within 30 days after discharge.

For CMU-MOSI and POM datasets, we follow the settings in [32] and adopt respectively the **movie sentiment analysis**, and **movie traits prediction** tasks. Detailed evaluation metrics of each task is introduced in the supplementary materials.

Table 2: Results in AUROC, AUPR and F1 with 95% confidence intervals on the MIMIC-III and MIMIC-IV datasets with *totally matched* modalities. The best results are highlighted in **bold**. Our proposed DPMM outperforms the baselines in all cases.

| Model | IHM | | | READM | | |
|---|---|---|---|---|---|---|
| | AUROC (↑) | AUPR (↑) | F1 (↑) | AUROC (↑) | AUPR (↑) | F1 (↑) |
| **MIMIC-III** | | | | | | |
| MMTM [26] | $0.805_{(0.761, 0.847)}$ | $0.394_{(0.305, 0.496)}$ | $0.386_{(0.316, 0.456)}$ | $0.744_{(0.698, 0.788)}$ | $0.397_{(0.322, 0.479)}$ | $0.418_{(0.355, 0.474)}$ |
| DAFT [38] | $0.807_{(0.760, 0.849)}$ | $0.404_{(0.316, 0.509)}$ | $0.414_{(0.340, 0.483)}$ | $0.701_{(0.651, 0.746)}$ | $0.364_{(0.293, 0.440)}$ | $0.364_{(0.302, 0.420)}$ |
| Unified [17] | $0.832_{(0.788, 0.873)}$ | $0.437_{(0.345, 0.547)}$ | $0.454_{(0.374, 0.535)}$ | $0.729_{(0.682, 0.775)}$ | $0.405_{(0.329, 0.494)}$ | $0.410_{(0.345, 0.471)}$ |
| MedFuse [18] | $0.830_{(0.785, 0.870)}$ | $0.439_{(0.342, 0.540)}$ | $0.489_{(0.415, 0.559)}$ | $0.719_{(0.673, 0.764)}$ | $0.382_{(0.309, 0.465)}$ | $0.382_{(0.317, 0.443)}$ |
| DrFuse [54] | $0.835_{(0.791, 0.875)}$ | $0.511_{(0.415, 0.611)}$ | $0.463_{(0.388, 0.531)}$ | $0.727_{(0.679, 0.771)}$ | $0.418_{(0.342, 0.503)}$ | $0.437_{(0.361, 0.507)}$ |
| DPMM | $\mathbf{0.854}_{(0.829, 0.904)}$ | $\mathbf{0.532}_{(0.428, 0.635)}$ | $\mathbf{0.490}_{(0.409, 0.579)}$ | $\mathbf{0.754}_{(0.731, 0.774)}$ | $\mathbf{0.460}_{(0.380, 0.542)}$ | $\mathbf{0.461}_{(0.399, 0.523)}$ |
| **MIMIC-IV** | | | | | | |
| MMTM [26] | $0.799_{(0.763, 0.833)}$ | $0.421_{(0.356, 0.505)}$ | $0.456_{(0.403, 0.506)}$ | $0.705_{(0.667, 0.740)}$ | $0.423_{(0.362, 0.485)}$ | $0.463_{(0.411, 0.513)}$ |
| DAFT [38] | $0.815_{(0.782, 0.849)}$ | $0.443_{(0.378, 0.527)}$ | $0.462_{(0.404, 0.519)}$ | $0.731_{(0.696, 0.764)}$ | $0.423_{(0.369, 0.493)}$ | $0.486_{(0.439, 0.535)}$ |
| Unified [17] | $0.806_{(0.772, 0.836)}$ | $0.440_{(0.369, 0.524)}$ | $0.469_{(0.415, 0.518)}$ | $0.719_{(0.684, 0.754)}$ | $0.421_{(0.364, 0.494)}$ | $0.472_{(0.423, 0.518)}$ |
| MedFuse [18] | $0.809_{(0.776, 0.841)}$ | $0.434_{(0.367, 0.512)}$ | $0.466_{(0.416, 0.516)}$ | $0.717_{(0.682, 0.754)}$ | $0.424_{(0.363, 0.491)}$ | $0.456_{(0.406, 0.506)}$ |
| DrFuse [54] | $0.816_{(0.784, 0.846)}$ | $0.446_{(0.377, 0.525)}$ | $0.457_{(0.403, 0.510)}$ | $0.727_{(0.694, 0.760)}$ | $0.419_{(0.361, 0.482)}$ | $0.471_{(0.422, 0.515)}$ |
| DPMM | $\mathbf{0.826}_{(0.782, 0.850)}$ | $\mathbf{0.482}_{(0.439, 0.542)}$ | $\mathbf{0.526}_{(0.420, 0.534)}$ | $\mathbf{0.736}_{(0.704, 0.773)}$ | $\mathbf{0.455}_{(0.404, 0.529)}$ | $\mathbf{0.488}_{(0.394, 0.524)}$ |

Table 3: Results in AUROC, AUPR, and F1 with 95% confidence intervals on MIMIC-III and MIMIC-IV datasets with *partially matched* modalities (i.e., missing modalities). The best results are highlighted in **boldface**. Our proposed method DPMM outperforms the baselines in all cases.

| Model | IHM | | | READM | | |
|---|---|---|---|---|---|---|
| | AUROC (↑) | AUPR (↑) | F1 (↑) | AUROC (↑) | AUPR (↑) | F1 (↑) |
| **MIMIC-III** | | | | | | |
| MMTM [26] | $0.847_{(0.826, 0.866)}$ | $0.441_{(0.390, 0.502)}$ | $0.476_{(0.433, 0.518)}$ | $0.751_{(0.727, 0.776)}$ | $0.446_{(0.405, 0.491)}$ | $0.441_{(0.407, 0.473)}$ |
| DAFT [38] | $0.854_{(0.835, 0.873)}$ | $0.482_{(0.429, 0.539)}$ | $0.496_{(0.453, 0.537)}$ | $0.753_{(0.731, 0.775)}$ | $0.445_{(0.402, 0.490)}$ | $0.430_{(0.400, 0.463)}$ |
| Unified [17] | $0.849_{(0.829, 0.869)}$ | $0.491_{(0.443, 0.542)}$ | $0.481_{(0.442, 0.520)}$ | $0.754_{(0.729, 0.775)}$ | $\mathbf{0.448}_{(0.408, 0.491)}$ | $0.435_{(0.402, 0.468)}$ |
| MedFuse [18] | $0.850_{(0.829, 0.869)}$ | $0.480_{(0.432, 0.534)}$ | $0.482_{(0.441, 0.523)}$ | $0.754_{(0.730, 0.775)}$ | $0.444_{(0.404, 0.489)}$ | $0.439_{(0.403, 0.473)}$ |
| DrFuse [54] | $0.839_{(0.817, 0.860)}$ | $0.474_{(0.424, 0.529)}$ | $0.482_{(0.441, 0.524)}$ | $0.745_{(0.720, 0.766)}$ | $0.431_{(0.388, 0.472)}$ | $0.446_{(0.411, 0.479)}$ |
| DPMM | $\mathbf{0.860}_{(0.840, 0.878)}$ | $\mathbf{0.511}_{(0.462, 0.566)}$ | $\mathbf{0.498}_{(0.456, 0.543)}$ | $\mathbf{0.756}_{(0.734, 0.780)}$ | $0.445_{(0.358, 0.523)}$ | $\mathbf{0.460}_{(0.428, 0.493)}$ |
| **MIMIC-IV** | | | | | | |
| MMTM [26] | $0.853_{(0.838, 0.867)}$ | $0.508_{(0.469, 0.551)}$ | $0.483_{(0.451, 0.518)}$ | $0.763_{(0.746, 0.780)}$ | $0.470_{(0.438, 0.504)}$ | $0.463_{(0.437, 0.490)}$ |
| DAFT [38] | $0.858_{(0.844, 0.872)}$ | $0.524_{(0.487, 0.564)}$ | $0.495_{(0.467, 0.524)}$ | $0.761_{(0.743, 0.779)}$ | $0.464_{(0.432, 0.500)}$ | $0.469_{(0.442, 0.496)}$ |
| Unified [17] | $0.852_{(0.838, 0.867)}$ | $0.503_{(0.460, 0.542)}$ | $0.497_{(0.467, 0.527)}$ | $0.755_{(0.736, 0.774)}$ | $0.469_{(0.436, 0.503)}$ | $0.460_{(0.434, 0.487)}$ |
| MedFuse [18] | $0.855_{(0.841, 0.870)}$ | $0.506_{(0.467, 0.546)}$ | $0.486_{(0.454, 0.517)}$ | $0.753_{(0.735, 0.771)}$ | $0.459_{(0.426, 0.493)}$ | $0.456_{(0.432, 0.478)}$ |
| DrFuse [54] | $0.854_{(0.839, 0.868)}$ | $0.510_{(0.473, 0.551)}$ | $0.496_{(0.465, 0.528)}$ | $0.765_{(0.746, 0.781)}$ | $0.478_{(0.444, 0.510)}$ | $0.466_{(0.438, 0.492)}$ |
| DPMM | $\mathbf{0.860}_{(0.847, 0.877)}$ | $\mathbf{0.529}_{(0.480, 0.563)}$ | $\mathbf{0.518}_{(0.487, 0.547)}$ | $\mathbf{0.771}_{(0.752, 0.788)}$ | $\mathbf{0.486}_{(0.452, 0.518)}$ | $\mathbf{0.472}_{(0.445, 0.497)}$ |

Table 4: Results in AUROC, AUPR on the MIMIC-IV Readmission task with different combinations of modalities.

| Modality | MMTM [26] | | DAFT [38] | | Unified [17] | | MedFuse [18] | | DrFuse [54] | | DPMM | |
|---|---|---|---|---|---|---|---|---|---|---|---|---|
| | AUROC | AUPR | AUROC | AUPR | AUROC | AUPR | AUROC | AUPR | AUROC | AUPR | AUROC | AUPR |
| T+I | 0.705 | 0.423 | 0.731 | 0.423 | 0.719 | 0.421 | 0.717 | 0.424 | 0.727 | 0.419 | 0.736 | 0.455 |
| T+N | 0.703 | 0.415 | 0.696 | 0.391 | 0.708 | 0.422 | 0.707 | 0.417 | 0.721 | 0.426 | 0.728 | 0.432 |
| T+I+N | 0.708 | 0.417 | 0.703 | 0.399 | 0.710 | 0.422 | 0.712 | 0.421 | 0.724 | 0.434 | 0.737 | 0.455 |
| | *T* only: 0.703 (AUROC), 0.420 (AUPR); | | | *I* only: 0.626 (AUROC), 0.296 (AUPR); | | | | *N* only: 0.538 (AUROC), 0.232 (AUPR) | | | | |

**Backbone Encoders.** We utilize ResNet34 [19] as the backbone encoder for CXR image data, following [18]. A two-layer stacked LSTM network [14] is employed for time-series data, including lab values and vital signs. For clinical notes and radiology reports, TinyBERT [25] serves as the encoder. Additionally, a projection layer maps all modality embeddings into the same latent space. For experiments on CMU-MOSI and POM datasets, we use an MLP as an encoder for each modality following [32].

## 4.2 Compared Methods

We compare DPMM on the clinical benchmark with five baselines: (1) **DrFuse** [54] adopts a transformer–type design which leverages disentangled representation learning to create a shared representation between the EHR and image modalities, even when one modality is missing. (2) **MMTM** [26] is a flexible plug-in module that facilitates information exchange between modalities. Since the model assumes availability of all modalities, we compensate for missing CXR and clinical notes during training and testing by filling in all zeros for missing values. (3) **DAFT** [38] is a module designed to exchange information between tabular data and image modalities when integrated into CNN models. Similarly, we replace missing CXR and clinical notes with zero matrices during training and testing. (4) **Unified** [17] is a dynamic approach for integrating auxiliary data modalities, learning modality-specific representations, and combining them via a unified classifier. It handles missing data inherently and leverages all available modality-specific information. (5) **MedFUSE** [18] employs LSTM-based fusion to combine features from image or language encoders with EHR encoders. It handles missing modalities by learning a global representation for absent CXR or clinical notes.

In addition to adapting the aforementioned baselines, we also incorporate two general multimodal baselines for CUM-MOSI and POM: **Low-rank Multimodal Fusion (LMF)** [32] adopts low-rank tensor to improve the efficiency of multimodal fusion; **Tensor Fusion Network (TFN)** [57] formulates the multimodal problem as modelling the intra-modal and inter-modal dynamics, and learns these two dynamics with learnable tensors.

### 4.3 Experimental Results

**Quantitative Results on MIMIC Datasets.** Tables 2 and 3 present the performance of DPMM compared to existing state-of-the-art (SOTA) methods on the MIMIC-III and MIMIC-IV datasets. The results demonstrate that our method can overall outperform the SOTA methods satisfactorily. Notably, for the IHM task, DPMM exceeds the best baseline by 2.3% in AUROC on MIMIC-III and 8% in AUPR on MIMIC-IV. Moreover, we discover that our method also performs satisfactorily on partially matched modalities, indicating the capability of the DP assumption to generate representations for missing modalities. We also perform experiments on tri-modalities with $M = 3$. Table 4 presents the results compared to SOTA methods. We observe in general an improved performance when more modalities are involved as more information can be gained. However, since the alignment complexity increases drastically with the increase in the number of modalities, the learning performance may not constantly outperform the one using fewer modalities.

**Quantitative Results on CMU-MOSI and POM.**

We further perform experiments CMU-MOSI and POM, where we follow the implementations by Liu et al. [32] [3]. Table 5 presents the results of sentiment analysis and movie trait prediction tasks where the modalities are video, audio, and text. We adapted the implementations of the clinical baselines to these datasets. Our method not only achieves competitive performance against the clinical baselines,

Table 5: Performance (%) comparison on two general multimodal datasets—CMU-MOSI, and POM.

| Model | CMU-MOSI | | | POM | | |
|---|---|---|---|---|---|---|
| | MAE (↓) | Accuracy (↑) | F1 (↑) | MAE (↓) | Corr (↑) | Accuracy (↑) |
| Unified [38] | 1.21 | 0.656 | 0.657 | 0.862 | 0.213 | 0.353 |
| MedFuse [18] | 1.11 | 0.700 | 0.696 | 0.861 | 0.262 | 0.334 |
| DrFuse [54] | 1.12 | 0.700 | 0.700 | 0.869 | 0.243 | 0.338 |
| LMF [32] | 1.13 | 0.697 | 0.698 | 0.856 | 0.266 | 0.343 |
| TFN [57] | 1.18 | 0.682 | 0.682 | 0.858 | 0.263 | 0.358 |
| DPMM | **1.09** | **0.701** | **0.702** | **0.847** | **0.273** | **0.359** |

but also some general multimodal learning baselines. This indicates that our model can also obtain a satisfactory performance on natural multimodal datasets in addition to the clinical context.

**Qualitative Results.** We visualize the capability of DPMM in selecting the most prominent features for every modality. Figure 5 presents the marginal distributions of the first mixture component of the $m$-th modality. We observe that the first components of each modality highlight the most significant features (with higher log-likelihood), while the second components are allocated less weights as their features are less prominent. This validates mixing different mixture components across modalities allows DP to select the most prominent feature distributions via its richer-gets-richer property.

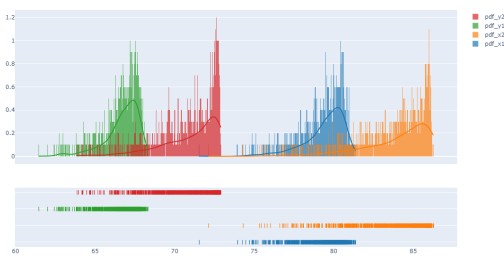

Figure 5: Histogram of the log-density values of the top 2 mixtures of each of the $M$ modalities. We observe that DP can rank the feature components of each modality according to its prominence.

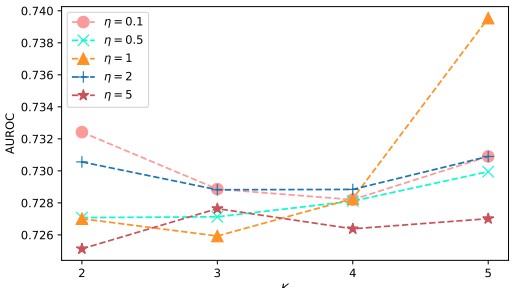

Figure 6: Performance in AUROC under the readmission task from MIMIC-IV with respect to different values of $K$ and $\eta$. Each line represents a distinct value of $\eta$.

### 4.4 Ablation Analysis

**Effect of Concentration Rate $\eta$ and Maximum Number of Mixture Components $K$.** The concentration rate $\eta$ determines the greediness of the DP in exploring new mixture components. The

---

[3] `https://github.com/declare-lab/multimodal-deep-learning/tree/main`

Table 6: Ablation study on different components (e.g., gradient-preserving sampling (GPS) and fusion module) of our proposed method. All results are reported in AUROC, AUPR, and F1 with 95% confidence intervals on the MIMIC-IV dataset.

| Model | Matched | IHM | | | READM | | |
|---|---|---|---|---|---|---|---|
| | | AUROC (↑) | AUPR (↑) | F1 (↑) | AUROC (↑) | AUPR (↑) | F1 (↑) |
| w/o alignment | × | $0.855_{(0.841, 0.870)}$ | $0.506_{(0.467, 0.546)}$ | $0.486_{(0.454, 0.517)}$ | $0.753_{(0.735, 0.771)}$ | $0.459_{(0.426, 0.493)}$ | $0.456_{(0.432, 0.478)}$ |
| w/o GPS | × | $0.855_{(0.840, 0.870)}$ | $0.521_{(0.484, 0.560)}$ | $0.489_{(0.456, 0.520)}$ | $0.761_{(0.744, 0.778)}$ | $0.466_{(0.432, 0.502)}$ | $0.467_{(0.444, 0.491)}$ |
| w/o fusion | × | $0.857_{(0.842, 0.872)}$ | $0.527_{(0.491, 0.568)}$ | $0.488_{(0.457, 0.522)}$ | $0.760_{(0.742, 0.779)}$ | $0.460_{(0.427, 0.493)}$ | $0.464_{(0.437, 0.490)}$ |
| DPMM | × | $\mathbf{0.860}_{(0.847, 0.877)}$ | $\mathbf{0.529}_{(0.489, 0.518)}$ | $\mathbf{0.518}_{(0.487, 0.547)}$ | $\mathbf{0.771}_{(0.752, 0.788)}$ | $\mathbf{0.486}_{(0.452, 0.518)}$ | $\mathbf{0.472}_{(0.445, 0.497)}$ |
| w/o alignment | ✓ | $0.809_{(0.776, 0.841)}$ | $0.434_{(0.367, 0.512)}$ | $0.466_{(0.416, 0.516)}$ | $0.717_{(0.682, 0.754)}$ | $0.424_{(0.363, 0.491)}$ | $0.456_{(0.406, 0.506)}$ |
| w/o fusion | ✓ | $0.807_{(0.774, 0.839)}$ | $0.436_{(0.367, 0.512)}$ | $0.456_{(0.399, 0.511)}$ | $0.711_{(0.676, 0.747)}$ | $0.414_{(0.355, 0.483)}$ | $0.447_{(0.394, 0.496)}$ |
| DPMM | ✓ | $\mathbf{0.826}_{(0.782,0.850)}$ | $\mathbf{0.482}_{(0.439,0.542)}$ | $\mathbf{0.526}_{(0.420, 0.534)}$ | $\mathbf{0.736}_{(0.704, 0.773)}$ | $\mathbf{0.455}_{(0.404, 0.529)}$ | $\mathbf{0.488}_{(0.394, 0.524)}$ |

Table 7: Ablation study on alignment losses. Results are reported in AUROC, AUPR, and F1 with 95% confidence intervals on the MIMIC-IV dataset.

| Alignment Loss | IHM | | | READM | | |
|---|---|---|---|---|---|---|
| | AUROC (↑) | AUPR (↑) | F1 (↑) | AUROC (↑) | AUPR (↑) | F1 (↑) |
| Cosine loss | $0.816_{(0.783, 0.849)}$ | $0.476_{(0.405, 0.560)}$ | $0.473_{(0.424, 0.521)}$ | $0.719_{(0.681, 0.756)}$ | $0.427_{(0.373, 0.493)}$ | $0.478_{(0.429, 0.525)}$ |
| KL loss | $0.820_{(0.788, 0.850)}$ | $0.464_{(0.392, 0.543)}$ | $0.499_{(0.439, 0.553)}$ | $0.724_{(0.685, 0.760)}$ | $0.436_{(0.379, 0.506)}$ | $0.458_{(0.414, 0.500)}$ |
| DPMM | $\mathbf{0.826}_{(0.782,0.850)}$ | $\mathbf{0.482}_{(0.439,0.542)}$ | $\mathbf{0.526}_{(0.420, 0.534)}$ | $\mathbf{0.736}_{(0.704, 0.773)}$ | $\mathbf{0.455}_{(0.404, 0.529)}$ | $\mathbf{0.488}_{(0.394, 0.524))}$ |

larger value is $\eta$, the faster the model concentrates to the significant features, while the parameter space would be less explored. On the other hand, employing a DP mixture model can inherently reduce the reliance on the number of mixtures as the DP can select the optimal number of mixtures by its stick-breaking process. Figure 6 demonstrates the performances in AUROC with respect to different combinations of $\eta$ and $K$. We discover that the performance is generally robust to different values of the two parameters, with a generally improved performance observed when the number of components is truncated at a high level (i.e., larger $K$).

**Effect of Proposed Components.** We analyze the contribution of each proposed component. Table 6 presents the performance with and without the fusion modules, and the gradient-preserving sampling module, respectively. We observe that on both the partially matched and completely matched datasets, there are performance improvements after applying these modules, validating their effectiveness.

**Effect of DP Alignment.** We also study the effects of the DP alignment loss, which is presented in Table 7. We observe that compared to commonly alignment losses (such as cosine similarity or empirical KL divergences), our method achieves a satisfactory performance.

# 5   Conclusion

We propose a deep DP-based framework for multimodal representation learning. We leverage the richer-gets-richer property of DP to amplify the features in each modality, while at the same time aligning their marginal distributions effectively. To deploy the DP model in high dimensions, we adopt variational inference to estimate the posterior distribution stochastically. Empirical analysis demonstrates the satisfactory performance of DPMM compared to SOTA methods, on both bimodal and trimodal datasets. Ablation analysis demonstrates the effectiveness of the DP mixture model in cross-modal alignment, and the robustness of the framework to changes in key hyperparameters.

**Limitations and Future Work.** There are other ways to apply Dirichlet process on multimodal learning, such as a hierarchical DP which models the mixture components and the marginal distributions through a hierarchy of Dirichlet processes. These potential variants will be explored in future works. More advanced assumptions, such as extending the diagonal covariance to full covariance and placing a Wishart prior to the covariance matrix, also warrant further exploration.

**Broad Impact.** DPMM offers a novel perspective for multimodal learning by dynamically allocating feature contributions and emphasizing prominent ones via its richer-gets-richer property, thereby enhancing feature fusion.

## Acknowledgements

This work was supported in part by the Research Grants Council of Hong Kong (27206123, 17200125, C5055-24G, and T45-401/22-N), the Hong Kong Innovation and Technology Fund (GHP/318/22GD), the National Natural Science Foundation of China (No. 62201483), and Guangdong Natural Science Fund (No. 2024A1515011875).

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

## Summary

In this supplementary material, we first present detailed information on the datasets in A.1 and tasks used in the experiments in A.2. Next, we provide more details on the implementation and hyperparameters used in the experiments in C.1 along with the settings of baseline methods in C.2. We present additional experimental results in D.

## A  Additional Information on Datasets and Tasks

### A.1  Datasets

Table A.1 provides a summary of the datasets used in our experiments.

**MIMIC-III** This dataset includes 46,520 ICU stays, each containing 17 clinical variables. Following the methodology in [15], the dataset is divided into training, validation, and test sets using a 70%-15%-15% split.

**MIMIC-IV** This dataset comprises 21,139 ICU stays and includes 17 clinical variables. Following [18], the data are divided into 70% for training, 10% for validation, and 20% for testing. For both the MIMIC-III and MIMIC-IV datasets, we extract 17 clinical variables that are routinely monitored in the ICU, consisting of 5 categorical and 12 continuous variables. In line with [18], the data is sampled every two hours during the first 48 hours of ICU admission for both tasks. This process results in a vector representation of size 76 at each time step in the clinical time-series data.

**MIMIC-CXR** This dataset comprises 377,110 chest X-ray images, 5,931 of which are linked to MIMIC-IV ICU stays. The data are divided into 4,287 training samples, 465 validation samples, and 1,179 test samples. In line with [18], we select the most recent Anterior-Posterior (PA) projection chest X-ray and apply transformations to the images, resizing them to $224 \times 224$ pixels. Additionally, this dataset includes radiology reports, which consist of unstructured text data. We utilize the radiology reports from the MIMIC-CXR dataset when evaluating the performance of DPMM on the tri-modal setting. Since these reports do not contain mortality information, they help mitigate potential overfitting and shortcut learning. The unstructured radiology reports are divided into four sections: Impression, Findings, Last Paragraph, and Comparison.

**MIMIC-III NOTE** This dataset contains 5,273 clinical notes linked to MIMIC-III ICU stays. The data are split into 3,652 samples for training, 815 for validation, and 806 for testing. Following the approach in [61], we use the last five clinical notes prior to the prediction time. If fewer than five notes are available for a given ICU stay, those notes are treated as missing. The initial number of matched ICU stays is approximately 15,000. To create the training, validation, and test sets, we randomly sample one-third of these ICU stays, ensuring that the scale of the clinical notes remains comparable to the CXRs in the MIMIC-IV dataset. Both the radiology report sections and clinical notes are limited to a maximum length of 512 words. These notes are tokenized into words and embedded into 312-dimensional vectors using the pre-trained TinyBERT model [25][4].

**CMU-MOSI** [58] This dataset comprises 93 opinion videos sourced from YouTube movie reviews. Each video is divided into multiple opinion segments, with each segment annotated for sentiment on a scale from -3 to 3, where -3 represents highly negative sentiment and 3 represents highly positive sentiment. Following the approach in [32], we partitioned the dataset into 1,283 samples for training, 229 for validation, and 686 for testing.

**POM** [37] This dataset consists of 903 videos containing movie reviews. Each video is labeled with speaker attributes, including: confident, passionate, voice, pleasant, dominant, credible, vivid, expertise, entertaining, reserved, trusting, relaxed, outgoing, thorough, nervous, persuasive, and humorous. In accordance with the method outlined in [32], we divided the dataset into 600 samples for training, 100 for validation, and 203 for testing.

---

[4]https://huggingface.co/huawei-noah/TinyBERT_General_4L_312D

Table A.1: Summary of real datasets used in the experiments.

| Dataset | Tasks | No. Train | No. Valid | No. Test | No. Pos. | Total |
|---|---|---|---|---|---|---|
| Complete Datasets | | | | | | |
| MIMIC-III | IHM | 14681 | 3222 | 3236 | 2795 | 21139 |
| MIMIC-III | READM | 14681 | 3222 | 3236 | 3987 | 21139 |
| MIMIC-III NOTE | – | 3652 | 815 | 806 | – | 5,273 |
| MIMIC-IV | IHM | 18064 | 2035 | 4972 | 3153 | 25071 |
| MIMIC-IV | READM | 18064 | 2035 | 4972 | 4603 | 25071 |
| MIMIC-CXR | – | 344529 | 9497 | 23069 | – | 377,095 |
| CMU-MOSI | – | 1,283 | 229 | 686 | – | 2,198 |
| POM | – | 600 | 100 | 203 | – | 903 |
| Matched Datasets | | | | | | |
| MIMIC-III & NOTE | IHM | 3652 | 815 | 806 | 736 | 5273 |
| MIMIC-III & NOTE | READM | 3652 | 815 | 806 | 998 | 5273 |
| MIMIC-IV & CXR | IHM | 4287 | 465 | 1179 | 890 | 5931 |
| MIMIC-IV & CXR | READM | 4287 | 465 | 1179 | 1262 | 5931 |
| MIMIC-IV & CXR & REPORT | IHM | 4287 | 465 | 1179 | 890 | 5931 |
| MIMIC-IV & CXR & REPORT | READM | 4287 | 465 | 1179 | 1262 | 5931 |

## A.2 Tasks

**In Hospital Mortality (IHM) Prediction.** The In Hospital Mortality (IHM) prediction task focuses on predicting whether a patient will pass away during his/her hospital stay. As summarized in Table A.1, the MIMIC-III dataset contains a total of 2,795 positive samples, of which 736 are matched with clinical notes. Similarly, the MIMIC-IV dataset includes 3,153 positive samples, with 890 matched to CXR.

**Readmission (READM) Prediction.** The Readmission (READM) prediction task aims to forecast whether a patient will be readmitted within 30 days of discharge. In this task, both patients who are readmitted and those who pass away in hospital are considered positive samples. As shown in Table A.1, the MIMIC-III dataset contains 3,987 positive samples, with 998 matched to clinical notes. In the MIMIC-IV dataset, there are 4,603 positive samples, with 1,262 matched to CXRs.

**Sentiment Analysis & Movie Traits Prediction.** We follow the implementations in [32] for the sentiment analysis and movie traits prediction tasks. We report the mean absolute error (MAE), accuracy, and F1 score for both tasks.

## B  Multivariate Gaussian Distribution

The multivariate Gaussian distribution is

$$p(\boldsymbol{z}; \boldsymbol{\mu}, \boldsymbol{\Sigma}) = \frac{1}{(2\pi)^{\frac{p}{2}} |\boldsymbol{\Sigma}|^{\frac{1}{2}}} \exp\left\{ -\frac{1}{2}(\boldsymbol{z} - \boldsymbol{\mu})^\top \boldsymbol{\Sigma}^{-1}(\boldsymbol{z} - \boldsymbol{\mu}) \right\},$$

where $\boldsymbol{\mu} \in \mathbb{R}^p$ is a $p$-dimensional mean vector and $\boldsymbol{\Sigma} \in \mathbb{R}^{p \times p}$ is the covariance matrix.

The KL divergence of two multivariate normal distributions $\mathcal{N}(\boldsymbol{\mu}_1, \boldsymbol{\Sigma}_1)$ and $\mathcal{N}(\boldsymbol{\mu}_2, \boldsymbol{\Sigma}_2)$ is

$$\mathrm{KL}(\mathcal{N}(\boldsymbol{\mu}_1, \boldsymbol{\Sigma}_1) \| \mathcal{N}(\boldsymbol{\mu}_2, \boldsymbol{\Sigma}_2)) = \frac{1}{2}\Big[ \log \frac{|\boldsymbol{\Sigma}_2|}{|\boldsymbol{\Sigma}_1|} - p + \mathrm{tr}\{\boldsymbol{\Sigma}_2^{-1}\boldsymbol{\Sigma}_1\} + (\boldsymbol{\mu}_2 - \boldsymbol{\mu}_1)^\top \boldsymbol{\Sigma}_2^{-1}(\boldsymbol{\mu}_2 - \boldsymbol{\mu}_1) \Big].$$

## C  More on Baseline Methods and Implementation Details

### C.1  Implementation Details and Hyperparameters

We train all models for 100 epochs using the training set, selecting the best-performing model based on validation AUROC. The final evaluation is conducted on the test set. The *Adam* optimizer is utilized for optimization, and early stopping is employed if the validation AUROC shows no

improvement over 15 consecutive epochs to mitigate overfitting. All experiments are performed on a single RTX-3090 GPU. The batch size is configured as 32 for the MIMIC-IV & CXR datasets and 16 for the MIMIC-III & NOTE datasets, except for DrFuse, which is trained with a batch size of 8. Hyperparameters are tuned using grid search on the validation set, and the test set results are based on the best configuration. The search space includes:

- Dropout ratio: $\{0, 0.1, 0.2, 0.3\}$
- Learning rate: $\{1 \times 10^{-4}, 5 \times 10^{-5}, 1 \times 10^{-5}\}$
- Concentration rate $\eta$: $\{0.1, 0.5, 1, 2, 5\}$
- Number of mixture components $K$: $\{2, 3, 4, 5\}$
- Temperature: $\{0.001, 0.005, 0.01, 0.05, 0.08\}$
- Regularization parameter $\lambda_{\text{DP}}$ (adjusting the strength of DP assumption): $\{1 \times 10^{-5}, 5 \times 10^{-6}, 1 \times 10^{-6}\}$

DPMM is developed using Python 3.11 and PyTorch 1.9. In line with MedFuse [18], ResNet34 [19] is employed as the backbone encoder for CXR images, a two-layer LSTM [14] is utilized for encoding time-series data, and pre-trained TinyBERT [25][5] is adopted for clinical notes. A projection layer aligns modality embeddings within the same latent space, followed by a two-layer LSTM as the fusion module to integrate embeddings. Finally, a multilayer perceptron (MLP) with a linear layer and a sigmoid activation function is used for classification.

## C.2 Additional Settings of Baseline Methods

We compare DPMM with the following baseline methods.

- **MMTM** [26] facilitates cross-modal information exchange through plugin module. Since it assumes all modalities are present, missing modalities (CXR and clinical notes) are filled with zeros during training and testing. For clinical notes, the ResNet34 encoder is replaced with TinyBERT for embedding.

- **DAFT** [38] integrates tabular and image data within CNN models. Similar to MMTM, missing modalities are replaced with zero matrices, and TinyBERT is used to embed clinical notes.

- **Unified** [17] is a dynamic method for integrating auxiliary data modalities by learning modality-specific representations and combining them through a unified classifier. It inherently manages missing modalities and leverages all available modality-specific data. TinyBERT is employed to embed clinical notes.

- **MedFuse** [18] is an LSTM-based fusion approach that combines features from image and EHR encoders. Missing modalities (CXR or clinical notes) are handled via global representations. Encoders for time-series data, clinical notes, and CXR are initialized randomly.

- **DrFuse** [54] employs disentangled representation learning to create a shared representation between EHR and image modalities, even with missing modalities. DrFuse uses ResNet50 as the image encoder and Transformer as the EHR encoder. For clinical notes, ResNet50 is replaced with TinyBERT.

- **Low-rank Multimodal Fusion (LMF)** [32] adopts low-rank tensor to improve the efficiency of multimodal fusion. We use an MLP as an encoder for each modality following [32].

- **Tensor Fusion Network (TFN)** [57] formulates the multimodal problem as modelling the intra-modal and inter-modal dynamics, and learns these two dynamics with learnable tensors. We use an MLP as an encoder for each modality following [32].

The implementation of DrFuse follows the original paper [54][6], and we use the same hyperparameters as the original paper. We directly adopt the implementations of MMTM, DAFT, Unified, and MedFuse

---

[5]`https://huggingface.co/huawei-noah/TinyBERT_General_4L_312D`
[6]`https://github.com/dorothy-yao/drfuse`

Table D.1: Results on using *learnable weights* for GMM on the MIMIC-IV dataset (partially / totally matched datasets).

| Model | IHM | | | READM | | |
|---|---|---|---|---|---|---|
| | AUROC | AUPR | F1 | AUROC | AUPR | F1 |
| learnable weights | 0.855 / 0.821 | 0.521 / 0.464 | 0.493 / 0.475 | 0.759 / 0.719 | 0.472 / 0.434 | 0.461 / 0.460 |
| DP weights | 0.860 / 0.826 | 0.529 / 0.482 | 0.518 / 0.526 | 0.771 / 0.736 | 0.486 / 0.455 | 0.472 / 0.488 |

provided by [18][7], and all hyperparameters are set to the default values provided by [18]. Moreover, the implementation of LMF and TFN follows [32][8], and we use the same hyperparameters as in the paper. We adapt the implementations of MMTM, DAFT, Unified, MedFuse and DrFuse to the tri-modal setting, including EHR time-series data, CXR images, and radiology reports.

# D    More Experimental Results

Table D.2: Results under different *missing ratios* of 10%, 40% and 70% on the READM task, MIMIC-IV dataset for different modalities (time series missing / image missing).

| Model | Matched | | 10% | | 40% | | 70% | |
|---|---|---|---|---|---|---|---|---|
| | AUROC | AUPR | AUROC | AUPR | AUROC | AUPR | AUROC | AUPR |
| MMTM | 0.705 | 0.423 | 0.695 / 0.704 | 0.417 / 0.422 | 0.658 / 0.703 | 0.356 / 0.420 | 0.589 / 0.699 | 0.297 / 0.400 |
| DAFT | 0.731 | 0.423 | 0.728 / 0.731 | 0.421 / 0.420 | 0.658 / 0.706 | 0.343 / 0.416 | 0.635 / 0.694 | 0.299 / 0.407 |
| Unified | 0.719 | 0.421 | 0.717 / 0.719 | 0.419 / 0.420 | 0.671 / 0.712 | 0.360 / 0.418 | 0.640 / 0.703 | 0.323 / 0.415 |
| MedFuse | 0.717 | 0.424 | 0.715 / 0.716 | 0.421 / 0.418 | 0.639 / 0.707 | 0.334 / 0.422 | 0.550 / 0.705 | 0.244 / 0.416 |
| DrFuse | 0.727 | 0.419 | 0.716 / 0.725 | 0.400 / 0.419 | 0.640 / 0.720 | 0.327 / 0.416 | 0.560 / 0.712 | 0.259 / 0.413 |
| DPMM | 0.736 | 0.455 | 0.735 / 0.731 | 0.442 / 0.439 | 0.675 / 0.716 | 0.367 / 0.426 | 0.649 / 0.714 | 0.331 / 0.423 |

## D.1    DP Weights against Learnable Weights

While learning weights through neural networks is viable, the selection of weights is observational. Therefore, we adopt DP as a probabilistic model to adjust for observational bias and leverage its richer-gets-richer property to select important features. We compare the performance of DPMM with learnable weights against DP weights on the MIMIC-IV dataset in Table D.1. The results show that using learnable weights might be biased and suboptimal, the ablation study (Tables 6–7 in the main text) also demonstrates the effectiveness of DP weights allocation.

## D.2    Different Missing Ratios

We define missing modality as the **observations** in each modality that is missing, leading to partially matched datasets. We adopt the common assumption that the missing mechanism is missing at random (MAR), the most commonly used one in statistics. We further validate that our model performs satisfactorily under different ratios of missingness. Table D.2 shows the results for different modalities with missing ratios of 10%, 40%, and 70% on the MIMIC-IV, READM tasks, where with more missingness, the performance deteriorates as expected. Moreover, the performance drop is more severe when the missingness is in the EHR time series modality, which indicates that the EHR is more informative for READM tasks.

## D.3    Comparison with More Imputation Methods

We compare our method with more recent and advanced imputation methods, such as PSA (Propensity Score Alignment) [51] and LSMT [28] on MIMIC-IV in Table D.3. Our method outperforms PSA and LSMT in both matched and partially matched settings.

---

[7] https://github.com/nyuad-cai/MedFuse
[8] https://github.com/Justin1904/Low-rank-Multimodal-Fusion

Table D.3: Performance comparison with PSA and LSMT.

| Match | Model | IHM | | READM | |
|-------|-------|-----|-----|-------|-----|
| | | AUROC | AUPR | AUROC | AUPR |
| Totally Matched | PSA [51] | 0.744 | 0.346 | 0.692 | 0.370 |
| | LSMT [28] | 0.803 | 0.444 | 0.701 | 0.421 |
| | DPMM | **0.826** | **0.482** | **0.736** | **0.455** |
| Partially Matched | PSA [51] | 0.792 | 0.386 | 0.720 | 0.391 |
| | LSMT [28] | 0.854 | 0.508 | 0.764 | 0.473 |
| | DPMM | **0.860** | **0.529** | **0.771** | **0.486** |

## D.4 Comparison with Contrastive Multimodal Models

We include comparisons with three representative explicit alignment models: CLIP [39], ALIGN [24], and CoMM [7], all evaluated under the totally matched setting on MIMIC4. As shown in Table D.4, our proposed method (DPMM) consistently outperforms these strong baselines across all metrics, demonstrating the effectiveness and competitiveness of our implicit alignment strategy in multimodal contrastive learning.

Table D.4: Performance comparison with contrastive multimodal models.

| Model | IHM | | READM | |
|-------|-----|-----|-------|-----|
| | AUROC | AUPR | AUROC | AUPR |
| CLIP [39] | 0.816 | 0.476 | 0.719 | 0.427 |
| ALIGN [24] | 0.819 | 0.469 | 0.725 | 0.428 |
| CoMM [7] | 0.823 | 0.465 | 0.726 | 0.432 |
| DPMM | **0.826** | **0.482** | **0.736** | **0.455** |

## D.5 Ablation on the Amplification and Alignment of DP

To disentangle the effects of DP-driven amplification and feature alignment, we conducted a controlled ablation study in Table D.5. To isolate the alignment effects, we train a DP Gaussian mixture model for each modality instead of mixing the mixture components together. The results show that both amplification and alignment contribute to performance gains, with the combination achieving the best results under both totally and partially matched settings. This demonstrates that DPMMbenefits from the dominant feature amplification as well as its implicit alignment across modalities.

## D.6 Analysis of the Computational Cost

We report the computational statistics (including the parameter size and training time) for all baselines and our DPMM model in Table D.6. DPMM incurs only slightly higher training time than other methods (e.g., 26.57s vs. 18–22s per epoch), which we consider acceptable given its modeling advantages. These results suggest that the stick-breaking process presents a reasonable computational bottleneck in standard multimodal learning scenarios.

Table D.5: Performance on the amplification and alignment of DP.

| Match | Model | IHM | | READM | |
|---|---|---|---|---|---|
| | | AUROC | AUPR | AUROC | AUPR |
| Totally Matched | w/o DP | 0.809 | 0.434 | 0.717 | 0.424 |
| | Amplification only | 0.816 | 0.462 | 0.724 | 0.429 |
| | Amplification + Alignment | **0.826** | **0.482** | **0.736** | **0.455** |
| Partially Matched | w/o DP | 0.855 | 0.506 | 0.753 | 0.459 |
| | Amplification only | 0.856 | 0.525 | 0.752 | 0.468 |
| | Amplification + Alignment | **0.860** | **0.529** | **0.771** | **0.486** |

Table D.6: Computational cost.

| Model | Batch size | No. Params | Training time per epoch (totally / partially matched) |
|---|---|---|---|
| MMTM [26] | 32 | 89.62 MB | 22.39s / 79.24s |
| DAFT [38] | 32 | 85.16 MB | 18.49s / 40.74s |
| Unified [17] | 32 | 87.72 MB | 18.97s / 41.58s |
| MedFuse [18] | 32 | 91.04 MB | 17.87s / 40.00s |
| DrFuse [54] | 32 | 200.92 MB | 18.73s / 61.43s |
| DPMM | 32 | 93.68 MB | 26.57s / 70.80s |

