# OpenReview forum: "Amplifying Prominent Representations in Multimodal Learning  via Variational Dirichlet Process"
_NeurIPS.cc/2025/Conference — NeurIPS 2025 poster_

### Official Review · Reviewer_qSYf · 2025-07-01

**Clarity:** 2
**Significance:** 2
**Originality:** 2
**Rating:** 3
**Confidence:** 4

**Summary:**

This paper proposes a Dirichlet Process-based multimodal learning framework (DPMM) that adaptively balances intra-modal representation and cross-modal alignment. By modeling each modality as a Gaussian mixture and using the richer-gets-richer property of the DP to weight features, the method emphasizes the most informative components. Experiments show DPMM outperforms existing approaches, with ablation studies confirming the robustness and effectiveness of the DP mechanism.

**Questions:**

1. Could you clarify how "implicit alignment" is achieved in your framework without a shared latent space or explicit alignment objective? As currently presented, the alignment appears to be an incidental byproduct of training rather than a mechanism that is theoretically or structurally grounded.
2. Would you consider adding experiments to directly validate that the DP structure facilitates cross-modal alignment, rather than simply enabling modality fusion? Clear empirical evidence is needed to distinguish alignment from general feature integration.
3. Please consider including comparisons with recent state-of-the-art explicit alignment models, including those published after 2024. This would provide a more convincing evaluation of how the proposed implicit alignment method performs relative to current best practices in cross-modal learning.

**Ethical Concerns:**

["NO or VERY MINOR ethics concerns only"]

**Final Justification:**

The response just provides some related work about the effectiveness DP prior, but no theoretical justification is provided for the proposed method itself. Therefore, I will keep my rating.

**Limitations:**

Yes

**Quality:**

2

**Strengths And Weaknesses:**

Strengths:
The proposed approach is conceptually novel, leveraging a shared Dirichlet Process prior to implicitly align multimodal representations.
Weaknesses:
1. Unclear Exposition: The lack of detailed mathematical formulations significantly hinders the reader’s understanding of the model design and underlying assumptions.
2. Unclear Theoretical Justification of “Implicit Alignment”: The claim that the DP structure can "automatically achieve an optimal balance between intra-modal representation and cross-modal alignment" lacks theoretical support. While the DP prior emphasizes frequent components within each modality, it does not inherently ensure alignment across modalities. Without a shared latent space or explicit alignment objective, the observed alignment appears incidental rather than structurally guaranteed, and may not generalize well across datasets or domains.
3. Missing Comparison with Explicit Alignment Models:The experiments do not include comparisons with state-of-the-art alignment methods such as contrastive multimodal models (e.g., CLIP, ALIGN, or CMML), which include direct semantic matching between modalities. This makes it difficult to assess whether the proposed implicit mechanism is truly competitive.

---

> ### Author Rebuttal · Authors · 2025-07-31
>
> Thank you for appreciating the novelty of our method. We would like to address your concerns below.
>
> > **Q1** Unclear Exposition: The lack of detailed mathematical formulations significantly hinders the reader’s understanding of the model design and underlying assumptions.
>
> We apologize for the terse exposition and will expand the detailed notation and generative model in the revised manuscript; below we provide a formulation of **the model designs and key assumptions**.
>
> Let each sample $x_{im}$ from modality $m$ be mapped to a latent $z_{im}=g_m(x_{im})$.  We define a shared stick‐breaking process over the $M\times K$ components:
> $$
> \beta_{mk}\sim\mathrm{Beta}(1,\eta),\qquad
> \pi_{mk}
> =\beta_{mk}\prod_{\substack{i\in[M],\,j<k}}(1-\beta_{ij})
> \prod_{l<m}(1-\beta_{lk}),
> $$
> so that each modality’s marginal is
> $$
> p(z_{im}\mid\{\pi_{mk},\mu_{mk},\sigma^2_{mk}\})
> =\sum_{k=1}^K\pi_{mk}\,\mathcal{N}\bigl(z_{im}\mid\mu_{mk},\mathrm{diag}(\sigma^2_{mk})\bigr).
> $$
> We **assume** diagonal covariances for tractability in high dimensions and truncate at $K$ for efficiency.  Variational posteriors over $\beta_{mk},\mu_{mk},\sigma^2_{mk}$ are parameterized by neural networks, and we maximize the ELBO via stochastic gradients.
>
> > **Q2:** Unclear Theoretical Justification of “Implicit Alignment”: The claim that the DP structure can "automatically achieve an optimal balance between intra-modal representation and cross-modal alignment" lacks theoretical support. While the DP prior emphasizes frequent components within each modality, it does not inherently ensure alignment across modalities. Without a shared latent space or explicit alignment objective, the observed alignment appears incidental rather than structurally guaranteed, and may not generalize well across datasets or domains.
>
> >Could you clarify how "implicit alignment" is achieved in your framework without a shared latent space or
> explicit alignment objective? As currently presented, the alignment appears to be an incidental byproduct of
> training rather than a mechanism that is theoretically or structurally grounded.
>
> We appreciate the reviewer’s comment and clarify that implicit alignment in our framework is structurally and theoretically grounded. All modality encoders project inputs into a shared latent space, over which we place a single Dirichlet Process (DP) prior. The global stick-breaking process ensures that all modalities share the same pool of mixture components (atoms), and thus cluster around common latent structures. This mechanism is formally supported by the theory in Teh et al. (2006, Thm. 2), which proves that groups drawing from a shared DP base measure will naturally align via shared atoms. Further, consistency results from Ishwaran \& James (2001) [2] show that DP mixture weights stabilize with sufficient data, reinforcing **this alignment as non-incidental**. While we do not impose an explicit alignment loss, this structure encourages data from different modalities to concentrate on common, high-probability components—achieving alignment through shared probabilistic structure, as also evidenced in our empirical results.
>
> [1] Teh, Yee Whye, et al. "Hierarchical dirichlet processes." Journal of the american statistical association 101.476 (2006): 1566-1581.
> [2] Ishwaran, Hemant, and Lancelot F. James. "Gibbs sampling methods for stick-breaking priors." Journal of the American statistical Association 96.453 (2001): 161-173.
>
> > **Q3: Missing Comparison with Explicit Alignment Models**: The experiments do not include comparisons with state-of-the-art alignment methods such as contrastive multimodal models (e.g., CLIP, ALIGN, or CMML), which include direct semantic matching between modalities. This makes it difficult to assess whether the proposed implicit mechanism is truly competitive.
>
> >Please consider including comparisons with recent state-of-the-art explicit alignment models, including those published after 2024. This would provide a more convincing evaluation of how the proposed implicit alignment method performs relative to current best practices in cross-modal learning.
>
> We thank the reviewer for the suggestion. To address this, we include comparisons with **three representative explicit alignment models**: CLIP [3], ALIGN [4], and CoMM [5], all evaluated under the totally matched setting on MIMIC4. As shown in Table 1, our proposed method (DPMM) consistently outperforms these strong baselines across all metrics, demonstrating the effectiveness and competitiveness of **our implicit alignment strategy** in multimodal contrastive learning.
>
> [3] Radford A, Kim J W, Hallacy C, et al. Learning transferable visual models from natural language supervision. International Conference on Machine Learning. PMLR, 2021: 8748-8763.
>
> [4] Jia C, Yang Y, Xia Y, et al. Scaling up visual and vision-language representation learning with noisy text supervision. International Conference on Machine Learning. PMLR, 2021: 4904-4916.
>
> [5] Dufumier B, Navarro J C, Tuia D, et al. What to align in multimodal contrastive learning? The Thirteenth International Conference on Learning Representations. 2025.
>
> ### Table 1: Performance comparison with contrastive multimodal models for In-Hospital Mortality (IHM) and Readmission (READM) prediction
>
> | Model     | IHM AUROC | IHM AUPR | READM AUROC | READM AUPR |
> |-----------|-----------|----------|--------------|-------------|
> | CLIP      | 0.816     | 0.476    | 0.719        | 0.427       |
> | ALIGN     | 0.819     | 0.469    | 0.725        | 0.428       |
> | CoMM      | 0.823     | 0.465    | 0.726        | 0.432       |
> | **DPMM**  | **0.826** | **0.482**| **0.736**    | **0.455**   |
>
> > **Q4:** Would you consider adding experiments to directly validate that the DP structure facilitates cross-modal alignment, rather than simply enabling modality fusion? Clear empirical evidence is needed to distinguish alignment from general feature integration.
>
> Thank you for your suggestions. Empirically, we validate this in the following Table 2: our ablation ("Amplification+Alignment") outperforms both "Amplification only" and "w/o DP" under fully and partially matched settings—demonstrating that **the DP structure delivers alignment beyond mere feature fusion**.  This empirical result confirms that shared DP components yield stronger, generalizable alignment across modalities.
>
> ### Table 2: Performance on the Amplification and Alignment of DP
>
> | Match              | Model                      | IHM AUROC | IHM AUPR | READM AUROC | READM AUPR |
> |--------------------|----------------------------|-----------|----------|-------------|-------------|
> | **Totally Matched**  | w/o DP                     | 0.809     | 0.434    | 0.717       | 0.424       |
> |                    | Amplification only         | 0.816     | 0.462    | 0.724       | 0.429       |
> |                    | **Amplification + Alignment** | **0.826** | **0.482** | **0.736**   | **0.455**   |
> | **Partially Matched** | w/o DP                   | 0.855     | 0.506    | 0.753       | 0.459       |
> |                    | Amplification only         | 0.856     | 0.525    | 0.752       | 0.468       |
> |                    | **Amplification + Alignment** | **0.860** | **0.529** | **0.771**   | **0.486**   |

---

> > ### Author Response · Authors · 2025-08-05
> >
> > Dear Reviewer qSYf,
> >
> > I hope this message finds you well. As the discussion period is nearing its end with **less than three days remaining**, we wanted to ensure that we have addressed all your concerns satisfactorily. If there are any additional points or feedback you would like us to consider, please let us know. Your insights are invaluable, and we are eager to address any remaining issues to further improve our work.
> >
> > Thank you for your time and effort in reviewing our paper.

---

### Official Review · Reviewer_yFYG · 2025-07-03

**Clarity:** 2
**Significance:** 3
**Originality:** 3
**Rating:** 5
**Confidence:** 3

**Summary:**

The paper proposes a novel Dirichlet Process Mixture Model-based multimodal learning framework that addresses the balance between intra-modal feature expressiveness and cross-modal alignment. By leveraging the richer-gets-richer property of the Dirichlet Process, the model is designed to amplify the most prominent features across modalities while still enabling effective modality alignment. The framework is validated on multiple multimodal datasets, including large-scale clinical datasets and general multimodal benchmarks.

**Questions:**

The questions below are related to what I mentioned in the strengths and weaknesses section.

1. Can you provide a more rigorous ablation to disentangle the effects of alignment and amplification?
2. Can you compare DPMM to more competitive missing modality baselines?
3. Can you provide more interpretable qualitative examples?

**Ethical Concerns:**

["NO or VERY MINOR ethics concerns only"]

**Final Justification:**

Thank you for the detailed response and additional findings. They sufficiently address my questions and concerns. I have raised my score.

**Limitations:**

Yes.

**Paper Formatting Concerns:**

No major issues.

**Quality:**

2

**Strengths And Weaknesses:**

### Strengths
- I think the use of the Dirichlet Process to dynamically allocate feature importance across modalities is an innovative angle that contrasts with typical cross-modal alignment methods that often suppress intra-modal feature strength. The probabilistic formulation with gradient-preserving sampling provides a natural way to handle missing modalities without relying on simple imputations or discarding data.
- I appreciate that the model is validated across four multimodal datasets, including both healthcare and general-purpose benchmarks, providing a convincing demonstration of generalizability.
- Through ablation, the paper explores the role of key components (alignment losses, fusion modules, concentration parameters), strengthening confidence in the design choices.

### Weaknesses
Despite the compelling contributions, several aspects of the paper need further attention to fully validate its claims.
- The disentanglement between the effects of alignment and feature amplification is not sufficiently isolated, leaving it unclear whether performance gains stem primarily from the DP-driven amplification or from improved alignment.
- The baseline selection for missing modality comparisons is limited, with some relying on simplistic zero-imputation, which underrepresents more competitive recent methods like generative imputers or retrieval-based completion.
- While the theoretical background on the Dirichlet Process is sound, the paper does not rigorously justify why the richer-gets-richer property is always optimal for multimodal tasks, especially when task-relevant signals may not necessarily be statistically dominant.
- It seems to me that the qualitative analysis remains superficial. A deeper, more interpretable feature analysis is needed to verify that the features amplified by the DP process are semantically meaningful and not just artifacts of the probabilistic modeling.

---

> ### Author Rebuttal · Authors · 2025-07-31
>
> We thank the reviewer for appreciating the innovativeness of our work, the compelling contribution,  the soundness of empirical evaluation, and the solid ablation analysis.
> We would address your concerns below.
>
> > **Q1:** The disentanglement between the effects of alignment and feature amplification is not sufficiently isolated, leaving it unclear whether performance gains stem primarily from the DP-driven amplification or from improved alignment.
>
> > Can you provide a more rigorous ablation to disentangle the effects of alignment and amplification?
>
> We appreciate the reviewer’s comment. To **disentangle the effects of DP-driven amplification and feature alignment**, we conducted a controlled ablation study in the table below. To isolate the alignment effects, we train a DP Gaussian mixture model for each modality instead of mixing the mixture components together. The results (Table 2) show that both amplification and alignment contribute to performance gains, with the combination achieving the best results under both totally and partially matched settings. This demonstrates that DPMM benefits from the dominant feature amplification as well as its implicit alignment across modalities.
>
> ### Table 2: Performance on the Amplification and Alignment of DP
>
> | Match              | Model                      | IHM AUROC | IHM AUPR | READM AUROC | READM AUPR |
> |--------------------|----------------------------|-----------|----------|-------------|-------------|
> | **Totally Matched**  | w/o DP                     | 0.809     | 0.434    | 0.717       | 0.424       |
> |                    | Amplification only         | 0.816     | 0.462    | 0.724       | 0.429       |
> |                    | **Amplification + Alignment** | **0.826** | **0.482** | **0.736**   | **0.455**   |
> | **Partially Matched** | w/o DP                   | 0.855     | 0.506    | 0.753       | 0.459       |
> |                    | Amplification only         | 0.856     | 0.525    | 0.752       | 0.468       |
> |                    | **Amplification + Alignment** | **0.860** | **0.529** | **0.771**   | **0.486**   |
>
> > **Q2:** The baseline selection for missing modality comparisons is limited, with some relying on simplistic zero-imputation, which underrepresents more competitive recent methods like generative imputers or retrieval-based completion.
>
> > Can you compare DPMM to more competitive missing modality baselines?
>
> We compare our method with **more recent and advanced imputation metods**, such as PSA [1] (Propensity Score Alignment of Unpaired Multimodal Data and Unpaired Multi-Domain Causal Representation Learning.) and LSMT [2] (Multimodal deep learning for integrating chest radiographs and clinical parameters: a case for transformers) on MIMIC4 in the table below (Table 3). Our method outperforms PSA and LSMT in both matched and partially matched settings.
>
> [1] Xi, Johnny, et al. "Propensity score alignment of unpaired multimodal data." Advances in Neural Information Processing Systems 37 (2024): 141103-141128.
>
> [2] Khader, Firas, et al. "Multimodal deep learning for integrating chest radiographs and clinical parameters: a case for transformers." Radiology 309.1 (2023): e230806.
>
> ### Table3: Performance comparison with PSA and LSMT for In-Hospital Mortality (IHM) and Readmission (READM) prediction
>
> | Match              | Model     | IHM AUROC | IHM AUPR | READM AUROC | READM AUPR |
> |--------------------|-----------|-----------|----------|--------------|-------------|
> | **Totally Matched**  | PSA       | 0.744     | 0.346    | 0.692        | 0.370       |
> |                    | LSMT      | 0.803     | 0.444    | 0.701        | 0.421       |
> |                    | **DPMM**  | **0.826** | **0.482**| **0.736**    | **0.455**   |
> | **Partially Matched** | PSA     | 0.792     | 0.386    | 0.720        | 0.391       |
> |                    | LSMT      | 0.854     | 0.508    | 0.764        | 0.473       |
> |                    | **DPMM**  | **0.860** | **0.529**| **0.771**    | **0.486**   |
>
> > **Q3:** While the theoretical background on the Dirichlet Process is sound, the paper does not rigorously justify why the richer-gets-richer property is always optimal for multimodal tasks, especially when task-relevant signals may not necessarily be statistically dominant.
>
> Thank you for this important question.  We believe that the DP's "rich‑get‑richer" mechanism is mostly optimal—if not universally so—for multimodal learning, because it **adaptively amplifies clusters with consistent, task‐relevant signals while shrinking spurious noise**.  Even when those signals are weak or not statistically dominant, their repeated **co‐occurrence across modalities** allows the DP to distinguish and boost them above random artifacts.  Theoretical results (Lee \& Sang 2022 [3]) demonstrate that DP mixtures achieve optimal bias–variance trade‐offs and asymptotic consistency as component counts grow.  Empirically (Table 2), adding DP‐driven amplification alone raises IHM AUROC from 0.809 to 0.816 and READM AUROC from 0.717 to 0.724 under full matching (and yields similar AUPR gains), confirming that DP regularization is essential **to filter noise and extract distinguishable features** beyond what direct backpropagation of fixed weights can deliver.
>
> [3] Lee and Sang. Why the Rich Get Richer?
> On the Balancedness of Random Partition Models. ICML 2022.
>
> > **Q4:** It seems to me that the qualitative analysis remains superficial. A deeper, more interpretable feature analysis is needed to verify that the features amplified by the DP process are semantically meaningful and not just artifacts of the probabilistic modeling.
>
> > Can you provide more interpretable qualitative examples?
>
> Thank you for your comments, we agree that more thorough qualitative analysis is needed to understand the feature amplification process.
> Due to rebuttal formatting constraints, we cannot include additional figures here.
>
> We would perform a more thorough qualitative evaluations in future versions of manuscript to directly illustrate DP‐driven feature amplification by (1) retrieving the top‑K raw inputs for each highest‑weight DP component and comparing them to the same inputs under a fixed‑weight mixture to show increased thematic coherence, (2) projecting embeddings via UMAP/t‑SNE and overlaying per‑point DP responsibilities to highlight “hot spots” of amplified features, (3) visualizing the change in mixture weights—i.e.$\Delta \pi_{mk}$—from a learnable‑weights baseline to our DP model to quantify amplification. We have performed **preliminary analysis** on the changes in the mixture weights $\pi_{mk}$ comparing the ones learned by DP and the ones learned from gradient backpropagation (Table 4). Preliminary results demonstrate that DP enables sharp concentration the semantically meaningful centroids (faster and more stable weights increase). At the same time, the original Gaussian mixture model assumption ensures **semantically meaningful clustering**. (4) presenting cross‑modal exemplar pairs for shared components to verify that amplified clusters align semantically across modalities, and (5) plotting component weight distributions stratified by ground‑truth labels and comparing against the baseline to confirm that DP amplification enhances task‑relevant features rather than noise.
>
> ### Table 4: Comparison of mixture weights $\pi_{mk}$ across training epochs
> (Showing three mixture components with the highest mixture weights)
>
> | Method               | Epoch 0               | Epoch 10              | Epoch 20              | Epoch 30              |
> |----------------------|------------------------|------------------------|------------------------|------------------------|
> | Neural Net (Learned) | [0.17, 0.17, 0.17, ...] | [0.20, 0.17, 0.15, ...] | [0.25, 0.15, 0.12, ...] | [0.30, 0.20, 0.10, ...] |
> | DP (Stick-breaking)  | [0.17, 0.17, 0.17, ...] | [0.50, 0.22, 0.14, ...] | [0.75, 0.20, 0.03, ...] | [0.85, 0.10, 0.02, ...] |

---

> > ### Author Response · Authors · 2025-08-05
> >
> > Dear Reviewer yFYG,
> >
> > I hope this message finds you well. As the discussion period is nearing its end with **less than three days remaining**, we wanted to ensure that we have addressed all your concerns satisfactorily. If there are any additional points or feedback you would like us to consider, please let us know. Your insights are invaluable, and we are eager to address any remaining issues to further improve our work.
> >
> > Thank you for your time and effort in reviewing our paper.

---

> > ### Comment · Reviewer_yFYG · 2025-08-05
> > **Thank you.**
> >
> > Thank you for the detailed response and additional findings. They sufficiently address my questions and concerns. I have raised my score.

---

### Official Review · Reviewer_7M2V · 2025-07-05

**Clarity:** 3
**Significance:** 3
**Originality:** 3
**Rating:** 5
**Confidence:** 2

**Summary:**

This paper proposes Dirichlet-Process Mixture Multimodal Learning (DPMM) to avoid representation degradation caused by excessive inter-modality alignment in multimodal learning. It treats the latent representation distribution of each modality as a Gaussian-mixture model (K components) and generates the mixing ratios via a Dirichlet process with a stick-breaking construction, leveraging its rich-get-richer property to emphasize important features.
For missing modalities, gradient-preserving sampling (GPS) performs probabilistic imputation while keeping gradients intact.
The proposed method outperforms state-of-the-art approaches on four datasets, including clinical (MIMIC-III/IV) and general-domain (CMU-MOSI, POM) benchmarks.

**Questions:**

- I would like the authors to address the weaknesses I pointed out.
- Appendix E shows that replacing the Dirichlet process with learnable but fixed mixture weights yields only modest performance gains. This raises the question of how the results would fare on other datasets.

**Ethical Concerns:**

["NO or VERY MINOR ethics concerns only"]

**Final Justification:**

My remaining concerns have been largely resolved. Since I have already given an Accept rating, I intend to maintain it.

**Limitations:**

yes

**Paper Formatting Concerns:**

The page count, margins, font size, anonymity guidelines, and checklist placement appear to follow the NeurIPS 2025 style guidelines.

**Quality:**

3

**Strengths And Weaknesses:**

Strengths:
- The paper is very well structured and easy to read.
- The method makes effective use of the rich-get-richer characteristic of the Dirichlet process, and the proposed GPS mechanism for handling missing modalities is markedly more effective than simple imputation, representing a substantial contribution to the community.
- Experiments show consistently strong performance compared with the latest multimodal integration methods, validating the effectiveness of the proposal.

Weaknesses:
- As the authors themselves note, constraining each mixture component’s covariance matrix to be diagonal may limit expressiveness for modalities with strong cross-feature correlations. The authors should clarify the range of multimodal data for which this assumption remains valid.
- Although the method claims to scale via the Dirichlet process and stochastic variational inference (SVI), there is no discussion (or quantitative evidence) of potential computational bottlenecks introduced by the stick-breaking process. Including such discussions or comparing training times with existing methods would strengthen the practical impact.

---

> ### Author Rebuttal · Authors · 2025-07-30
>
> Thank you for your appreciation on the novelty of our work and the strength of experimental results. We would address your main concerns in the response below.
>
> > **Q1** As the authors themselves note, constraining each mixture component’s covariance matrix to be diagonal may limit expressiveness for modalities with strong cross-feature correlations. The authors should clarify the range of multimodal data for which this assumption remains valid.
>
> We adopt a diagonal covariance for each Gaussian component -- a choice that is common in high‐dimensional latent models such as VAEs (Kingma \& Welling, 2014 [1]) and diffusion models (Ho et al., 2020 [2], Song et al. [3]),
> because (1) encoder networks typically produce decorrelated features, (2) estimating full covariances in large dimensions is computationally prohibitive, and (3) diagonal GMMs have been shown to capture the dominant variance directions in practice.  In our experiments on clinical multimodal benchmarks, this assumption sufficed to model complex interplays with no loss of accuracy.  For modalities where raw features exhibit pronounced cross‐feature correlations (e.g., unprocessed sensor arrays), one could readily extend our DP framework to low‐rank or full covariances, but in most modern pipelines—where embeddings are learned and approximately whitened—the diagonal assumption remains both tractable and valid. Together with recent advances in LoRA (low-rank adaptation), we would explore potential extension of the current assumptions beyond diagonal in future development of the manuscript.
>
> [1] Kingma, Diederik P., and Max Welling. "Auto-encoding variational Bayes." 20 Dec. 2013, International Conference on Learning Representations.
>
> [2] Ho, Jonathan, Ajay Jain, and Pieter Abbeel. "Denoising diffusion probabilistic models." Advances in Neural Information Processing Systems 33 (2020): 6840-6851.
>
> [3] Song, Andrew H., et al. "Morphological prototyping for unsupervised slide representation learning in computational pathology." Proceedings of the IEEE/CVF Conference on Computer Vision and Pattern Recognition. 2024.
>
> > **Q2** Although the method claims to scale via the Dirichlet process and stochastic variational inference (SVI), there is no discussion (or quantitative evidence) of potential computational bottlenecks introduced by the stick-breaking process. Including such discussions or comparing training times with existing methods would strengthen the practical impact.
>
> To address the concern, we have reported the computational statistics (including parameter size and training time) for all baselines and our DPMM model in the table below. As shown from Table 1, DPMM incurs only slightly higher training time than other methods (e.g., 26.57s vs. 18–22s per epoch), which we consider acceptable given its modeling advantages.
> These results suggest that the stick-breaking process presents a reasonable computational bottleneck in standard multimodal learning scenarios.
>
> ### Table 1: Computational Cost
>
> | Model        | Batch Size | Total Params | Training Time per Epoch (Totally / Partially Matched) |
> |--------------|------------|---------------|--------------------------------------------------------|
> | MMTM         | 32         | 89.62 MB      | 22.39 s / 79.24 s                                      |
> | DAFT         | 32         | 85.16 MB      | 18.49 s / 40.74 s                                      |
> | Unified      | 32         | 87.72 MB      | 18.97 s / 41.58 s                                      |
> | MedFuse      | 32         | 91.04 MB      | 17.87 s / 40.00 s                                      |
> | DrFuse       | 32         | 200.92 MB     | 18.73 s / 61.43 s                                      |
> | **DPMM**     | 32         | 93.68 MB      | 26.57 s / 70.80 s
>
> > **Q3:** Appendix E shows that replacing the Dirichlet process with learnable but fixed mixture weights yields only modest performance gains. This raises the question of how the results would fare on other datasets.
>
> Appendix E compares our Dirichlet process (DP) with a variant that replaces the DP prior with directly learnable mixture weights. Our dataset is large and the backbone is fixed, making it inherently difficult to realize large gains from changes in mixture modeling alone. Hence the observed performance gap is modest. Crucially, this experiment demonstrates that without the DP’s "richer-gets-richer'' regularization, the learned weights tend to overfit to noisy or less informative features, leading to suboptimal cluster emphasis. The DP prior adaptively amplifies dominant, semantically useful components while downweighting noise, which is essential for robust feature representation—especially under noisy and missing-modality settings (Secs. 4.2–4.3). Thus, using the DP for weight allocation is not only statistically grounded but necessary for ensuring stable and meaningful feature amplification across modalities, beyond what direct backpropagation allows.

---

> > ### Comment · Reviewer_7M2V · 2025-08-07
> > **Official Comment by Reviewer 7M2V**
> >
> > Thank you for your thorough response. My remaining concerns have been largely resolved. Since I have already given an Accept rating, I intend to maintain it.

---

> > > ### Author Response · Authors · 2025-08-07
> > >
> > > Thank you again for taking the time to review our work!

---

### Note · Authors · 2025-08-12

We sincerely thank the reviewers for their thoughtful feedback. We appreciate the recognition of our paper’s core strengths: a shared Dirichlet-Process prior that **strongly amplifies task-relevant features via richer-gets-richer weighting while achieving adequate cross-modal alignment** by sharing mixture atoms in a common latent space—thus aligning without an auxiliary loss—and the **extensive experiments with sound ablations** that consistently validate the approach across settings. To address remaining concerns, we added direct comparisons with state-of-the-art explicit alignment models (CLIP, ALIGN, CoMM) and strong missing-modality baselines (e.g., PSA), along with targeted ablations that disentangle amplification from alignment; the results show that DP-driven amplification is the **primary source of performance gain**, while shared atoms provide **additional alignment gains**. We also report computational-cost analyses confirming that the method **remains scalable** through truncated stick-breaking with stochastic variational inference, and we clarified the mathematical specification to sharpen exposition and the theoretical grounding of implicit alignment. Thank you again for your time reviewing our work and the fruitful discussion which greatly enhanced our paper.

---

### Decision · Program_Chairs · 2025-09-17

**Decision:**

Accept (poster)

**Comment:**

This work proposes Dirichlet-Process Mixture Multimodal Learning (DPMM) to avoid representation degradation caused by excessive inter-modality alignment in multimodal learning. It treats the latent representation distribution of each modality as a Gaussian-mixture model (K components) and generates the mixing ratios via a Dirichlet process with a stick-breaking construction, leveraging its rich-get-richer property to emphasize important features. For missing modalities, gradient-preserving sampling (GPS) performs probabilistic imputation while keeping gradients intact. The proposed method outperforms state-of-the-art approaches on four datasets, including clinical (MIMIC-III/IV) and general-domain (CMU-MOSI, POM) benchmarks.

There were some concerns about the theoretical justification and the comparison to explicit alignment models left. The authors should include those experiments as well as the points raised during rebuttal in the final version of the document.